

# From apex to shoreline: fluvio-deltaic architecture for the Holocene Rhine-Meuse delta, The Netherlands

Marc J.P. Gouw[1], Marc P. Hijma[2]

[1]Utrecht University, Faculty of Geosciences, Department of Physical Geography, Heidelberglaan 2, P.O. Box 80115, 3508
TC Utrecht, The Netherlands
[2]Deltares Research Institute, Department of Applied Geology and Geophysics, Princetonlaan 6-8, P.O. Box 85467, 3508 AL
Utrecht, The Netherlands

*Correspondence to*: Marc Hijma (marc.hijma@deltares.nl)

**Abstract.** Despite extensive research on alluvial architecture, there is still a pressing need for data from modern fluvio-deltaic
environments. Previous research in the fluvial-dominated proximal and central Rhine-Meuse delta (The Netherlands) has
yielded clear spatial trends in alluvial architecture. In this paper, we include the backwater length to establish architectural
trends from apex to shoreline. Channel-belt sand body width/thickness ratios and interconnectedness were determined and the
proportions of fluvial channel-belt deposits, fluvial overbank deposits, organics and intertidal deposits were calculated for the
complete fluvio-deltaic wedge, based on high-resolution geological cross-sections. It was found that the average
width/thickness ratio of channel-belt sand bodies in the proximal delta is five times higher than in the distal delta. Other down-
valley trends include an 80%-decrease of the channel deposit proportion (CDP) and a near-constant proportion of overbank
deposits. Additionally, interconnectedness in the proximal delta is three times higher than in the distal delta. Based on the
Rhine-Meuse dataset, the authors propose a linear empirical function to model the spatial variability of CDP. It is argued that
this relationship is driven by four key factors that change along stream: channel lateral-migration rate, channel-belt longevity,
creation of accommodation space and inherited flood-plain width. Additionally, it is established that the sensitivity of CDP to
changes in the ratio between channel-belt sand body width and flood-plain with, (normalised channel-belt sand body width)
varies spatially and is greatest in the central and distal delta. Also, the proportion of fluvial channel-belt sands is generally an
appropriate proxy for the total sand content of fluvio-deltaic successions, albeit that its suitability as a total-sand indicator
rapidly fades in the distal delta. With this paper, unique high-resolution quantitative data and spatial trends on the alluvial
architecture are available for an entire delta, hereby providing a dataset that can be used to further improve existing fluvial
stratigraphy models.



## 1 Introduction

The architecture of fluvio-deltaic successions has been studied extensively in the past decades mainly because of the occurrence of valuable natural resources (water, hydrocarbons, precious metals) within strata of fluvio-deltaic origin (e.g., Tye et al., 1999; Ryseth, 2000; Kombrink et al., 2007). Alluvial architecture describes the proportion, distribution and geometry of fluvial sediment bodies in sedimentary basins (Allen, 1978). Alluvial-architecture studies primarily focus on the geometry of fluvial sand bodies (see Gibling, 2006 for an extensive overview), the proportion of channel-belt sands within fluvial successions

(e.g., Ryseth et al., 1998; Bridge et al., 2000; Flood and Hampson, 2015; Blum et al., 2013) and controlling factors reckoned to influence alluvial architecture (e.g., Heller and Paola, 1996; Zaleha, 1997; Hajek and Wolinsky, 2012). Despite the elaborate work done, there are still two aspects of interest concerning alluvial-architecture research.

First, the majority of published alluvial-architecture studies cover ancient (hard-rock) successions, often to provide analogues

for hydrocarbon-bearing formations (e.g., Aigner et al., 1996; Bridge et al., 2000; Dalrymple, 2001). A drawback of studying ancient successions is that they are typically incomplete and/or deformed due to post-depositional processes (erosion, faulting, compaction) which introduces significant uncertainty in the interpretation of alluvial architecture. Furthermore, time control of ancient formations is usually poor whereas adequate dating of the deposits is needed to characterize the architecture of a fluvial succession (Bridge, 2003). Moreover, acquired datasets of ancient deposits are mostly of limited (palaeo)geographic

extent. Most alluvial-architecture research of ancient formations therefore encompass merely a limited section of the larger fluvial system or delta the studied deposits are part of (e.g., Lopez-Gomez et al., 2009; Jensen and Pedersen, 2010; Corbett et al., 2011; York et al., 2011), meaning that the large-scale transition from river valley to coastal plain (and associated backwater effects) is understudied, while we know from modern systems that these effects are substantial (Blum et al., 2013; Wu and Nitterour, 2020). Comprehensive studies covering the alluvial architecture on a delta scale are consequently scarce (notable

exceptions include Hampson et al., 2012; Klausen et al., 2014; Lyster et al., In press)

The second point of interest in alluvial-architecture research is the constant need for – or chronic lack of – architectural field data to aid (and enhance) geological modelling. Modelling is commonly employed by (reservoir) geologists to reconstruct and understand three-dimensional fluvial stratigraphy, for example to support reservoir characterisation (see, e.g., Bridge, 2008;

Keogh et al., 2007). Geological models constantly need appropriate field data from the reservoir of interest, analogues and/or comparable modern fluvial systems to develop, test and/or improve the models. Particularly high-resolution data from which spatial trends can be deduced would be helpful to support geological modelling and to enhance fundamental understanding of the alluvial architecture of (ancient) fluvial and fluvio-deltaic successions.

The above-mentioned issues can be addressed by studying the architecture of modern records. The main advantage of studying modern fluvial successions is that they tend to be more-complete than their ancient counterparts The Holocene Rhine-Meuse



delta (fig. 1) is especially suitable for alluvial-architecture research because of the availability of a large amount of subsurface data (Berendsen et al., 2007; Cohen et al., 2012). The extensive Rhine-Meuse dataset includes lithological information from cores and cone penetration tests (CPTs), detailed geological maps and cross-sections, and $^{14}$C and other dates (OSL,

archaeological, historical). Furthermore, extensive research over the past decades has led to a sound understanding of the factors that controlled the development of the delta (see Törnqvist, 1994; Berendsen and Stouthamer, 2000; Cohen, 2005; Gouw and Erkens, 2007; Hijma et al., 2009; Stouthamer and Berendsen, 2000; Hijma and Cohen, 2011; Stouthamer et al., 2011), which can be used to explain alluvial architecture. Thus far, alluvial-architecture research in the Rhine-Meuse delta has concentrated on the fluvial-dominated upstream half of the delta (2008, 2007), i.e. the proximal and central delta (definition

cf. Stouthamer et al., 2011). For that area, Gouw (2008, 2007) reported a strong decrease in a downstream direction of both the proportion of fluvial channel-belt sands within the succession (channel deposit proportion, CDP) and the degree to which channel-belt sand bodies are interconnected (connectedness ratio, CR), largely because of the decrease of channel-belt sand body width relative to floodplain width. In this paper, new data from the lower reach of the Rhine-Meuse delta (distal delta) is evaluated, where marine and estuarine deposits intercalate with fluvial deposits, to test whether the calculated architectural

relationships from upstream still hold.

With this paper we aim to: (1) present new alluvial-architecture data from the distal Rhine-Meuse delta; (2) develop a function that describes the delta-wide spatial trend in CDP based on the Rhine-Meuse dataset; and (3) present new conceptual insights that can be used in other fluvio-deltaic settings as well.

**2 Geological setting**

The Rhine-Meuse delta is located in the south-east of the North Sea basin (Ziegler, 1994). To the north and south, undulating Pleistocene topography fringes the delta. The delta apex region is currently located 150-160 km upstream of the present coastline (Stouthamer et al., 2011), in the basin hinge zone. During the Quaternary, the Rhine and Meuse rivers repeatedly shifted their courses and main depocentres (e.g., Busschers et al., 2007; Hijma et al., 2012).

**2.1 Evolution and architecture of the Holocene fluvio-deltaic wedge**

Last Glacial and Early Holocene fluvial sediments underlie the Holocene fluvio-deltaic wedge. These coarse and gravelly sands have been deposited by precursors of the Rhine and Meuse rivers (Busschers et al., 2007; Hijma et al., 2009) (Kreftenheye Formation; table 1) and are capped by a characteristic floodplain loam with palaeosol development (Wijchen Member; Autin, 2008). Outside the palaeo-valley, aeolian sedimentation prevailed until the earliest Holocene (Boxtel

Formation). Relative sea-level rise after the Last Glacial Maximum caused onlap and the formation of the fluvio-deltaic wedge (or 'coastal prism' cf. Posamentier et al., 1992). The present Rhine-Meuse fluvio-deltaic wedge thickens in a western (downstream) direction to more than 20 m near the North Sea coast and is composed of a stacked succession of clastic fluvial,



estuarine and marine deposits, intercalated with organic layers (fig. 2). All Holocene clastic fluvial deposits in the fluvio-deltaic wedge belong to the Echteld Formation. This formation also includes freshwater estuarine deposits (Hijma et al., 2009).

Brackish estuarine and marine deposits are grouped in the Naaldwijk Formation. The organics are grouped in the separate Nieuwkoop Formation.

At the base of the Holocene, a near-continuous dm-thick peat layer is found which reflects drowning of the pre-Holocene topography and has been used to determine the timing of fluvial-deltaic onlap (e.g. Jelgersma, 1961; Hijma and Cohen, 2011;

Van De Plassche, 1982). Onlap started around 8.5 ka BP (note that all dates are in calendar years, unless stated otherwise) and was driven by high rates of sea-level rise that caused rapid drowning of the fluvial valley (Fig. 1; Hijma and Cohen, 2010; Hijma and Cohen, 2019) and the transformation to an estuary (Hijma and Cohen, 2011) at the start of the Middle Holocene. An up to 5 m thick layer of freshwater fluvial-tidal and bay-head delta deposits (sand and clay) in the downstream fluvio-deltaic wedge reflect these estuarine conditions. Further transgression caused sedimentation of marine intertidal deposits within

a back-barrier environment. These back-barrier sediments (an up to 5 m thick bed of fine sand and silty and sandy clay, mainly) reach up to ~40 km inland from the present coastline (figs. 1, 2). Under continuous but decelerating sea-level rise, fluvio-deltaic onlap progressively moved inland and by ~5 ka BP, net aggradation occurred practically all over the present delta (Berendsen and Stouthamer, 2000; Cohen et al., 2002; Stouthamer and Berendsen, 2000). The fluvial part of the wedge comprises numerous channel-belt sand bodies and associated overbank deposits. The sand bodies consist of fine to coarse

sand, sometimes admixed with gravel. The overbank deposits include natural levee, crevasse-splay, flood basin and lacustrine deposits. The natural levees (silty and sandy clay) fringe their associated channel-belt sand bodies in 50-500 m wide zones and have typical thicknesses of 1-3 m near the channel-belt edges, with 4-5 m as a maximum. The crevasse splays (sand, silty and sandy clay) cover an area of 10s-100s km² each. The splays are 1-2 m thick, whereas the infill of crevasse channels is typically twice as thick (Berendsen, 1982; Stouthamer, 2001; Weerts and Bierkens, 1993). Flood-basin deposits occur in up to 5 m thick

beds of massive clays. Additionally, flood-basin lake fills are found, on the distal delta plain mostly. The lakes existed in remote flood basins in-between the active river channels. The lake fills are partly organic, partly clastic ('organic-clastic lake fills' cf. Bos, 2010) and are deposited in up to 5 m thick beds. The lake sediments are largely of fluvial origin (Bos, 2010).

The clastic fluvio-deltaic deposits in the Rhine-Meuse delta are intercalated with organic beds. The organics mainly consist of

peat and form a significant part of the fluvio-deltaic wedge in the central and distal delta. Here, the organic beds are up to 7 m thick. In contrast, organics are virtually absent in the proximal delta; only thin (<1 m) isolated layers of peat are found here (fig. 2). Large-scale aggradation in the Rhine-Meuse delta ceased around 1200 AD (~0.75 ka BP) with the embankment of the Rhine-Meuse distributaries (Berendsen and Stouthamer, 2000). Presently, sedimentation only takes place in the embanked floodplains along the modern Rhine and Meuse distributary channels.




The Last Glacial and Early Holocene substrate of the Rhine-Meuse palaeovalley forms the foundation of the Holocene succession. This is the Lowstand Systems Tract (LST). Additionally, the earliest Holocene fluvial onlap sediments in the western delta are included in the LST ('pre-Transgressive Systems Tract (pre-TST)' cf. Cohen and Hijma, 2014). In the distal delta plain, the TST comprises brackish and freshwater estuarine sediments and intertidal deposits formed in a back-barrier

environment. Further upstream, fluvial, lacustrine and organic deposits formed in permanently inundated flood basins and extensive swamps are grouped in the TST. The upper boundary of the TST (or base of the Highstand Systems Tract, HST) in the distal delta is marked by the onset of widespread peat formation, dated at ~6 ka (Hijma and Cohen, 2011). The HST in the distal delta therefore largely consists of thick peat layers that have been partly mined. For the purpose of our calculations, the original succession was reconstructed, i.e. the excavated areas were artificially refilled. In the central delta, clastic fluvial

deposits and organics formed in periodically inundated flood basins and small-scale swamps are part of the HST. In the proximal delta, the HST directly overlies the LST. The TST is absent here because sediment delivery always outpaced accommodation-space creation in this part of the delta (Hijma and Cohen, 2011). The location of the downstream-most cross-section (H) marks the downstream end of our study area; shoreface and beach deposits and coastal dunes are not included in this study. See figure 1 for the location of the longitudinal sections.

**2.2 Cross sections**

Three transects illustrate the Holocene fluvio-deltaic succession of the Rhine-Meuse delta (fig. 3). These transects (see figure 1 for their location) are modified fragments of the cross-sections published earlier by Gouw and Erkens (2007, their cross-sections A-A' and D-D', also published in Gouw (2008)) and Hijma et al. (2009 their cross-section B-B'). Below, the characteristics of the Rhine-Meuse fluvio-deltaic architecture are briefly described with the presented transects as typical

examples for the preserved succession in the proximal (fig. 3*A*), central (fig. 3*B*) and distal (fig. 3*C*) delta. For the original cross-sections and elaborate descriptions thereof, we refer to the original papers (Gouw and Erkens, 2007; Hijma et al., 2009). The calculations of the alluvial-architecture parameters in this study are based on the full set of eight cross-sections as displayed in figure 1.

*Transect proximal delta*

The succession of the proximal delta is characterized by relatively wide channel-belt sand bodies, thin beds of overbank deposits and small-scale occurrence of organics (fig. 3*A*). The floodplain is fringed by Pleistocene uplands (fig. 1) and therefore of limited width (15-25 km). Gouw (2008) used this fact to explain the observed high interconnectedness in this part of the delta; most sand bodies are interconnected being up to 4300 m wide and 10 m thick. Additionally, all Holocene channel-belt

sand bodies are connected to the sandy Pleistocene substrate (LST). Up to 3-m-thick beds of overbank deposits bound the channel-belt sand bodies. The overbank deposits are underlain by a dm-thick peat layer, which reflects beginning of Holocene onlap.



*Transect central delta*

Contrary to upstream, the channel-belt sand bodies of the 50-60 km wide central delta are mostly not connected to another Holocene sand body, although a limited number of interconnected channel-belt sand bodies exist (fig. 3*B*). The isolated channel-belt sand bodies are 100-1,400 m wide and 5-7 m thick; the interconnected channel-belt sand bodies are up to 2,600 m wide and 11 m thick. The channel-belt sand bodies, ~75% of which make contact with the Pleistocene substrate, are encased in a 6-9 m thick succession of overbank fines and organics. The overbank beds within the succession are 1-4 m in thickness,

mostly. Organic layers are typically 0.5-2 m thick, although beds of 5 m thick are also found in areas with minimal fluvial activity during the course of the Holocene (see for example kms 25-27 in cross-section D-D' of Gouw and Erkens (2007) or cross-section D of Gouw (2008)).

*Transect distal delta*

The Holocene succession of the distal delta plain typically consists of narrow channel-belt sand bodies and thick beds of fluvial overbank deposits and organics (fig. 3*C*). Furthermore, estuarine and back-barrier intertidal deposits are found on the ~75 km wide distal delta plain. Most channel-belt sand bodies are not connected to another Holocene channel-belt sand body. However, practically all Holocene channel-belt sand bodies are attached to the underlying Pleistocene sands. Channel-belt sand body width is typically between 100 m and 600 m and ~1200 m at a maximum. The thickness of the isolated channel-belt sand

bodies ranges from 4 m to 12.5 m; that of interconnected Holocene channel-belt sand bodies is up to 17 m.

The sand bodies are encased in an up to 15 m thick succession of fluvial overbank fines, estuarine deposits, organics and intertidal back-barrier deposits. Most overbank deposits are encountered in the lower half of the succession, roughly below 6 m-OD. Hijma et al. (2009) reckoned that a large part of these sediments are estuarine in nature, deposited partly under freshwater and partly under brackish conditions. The estuarine deposits are overlain by a 1-3 m thick peat layer that is present

throughout the distal delta plain (e.g., Beets and Van Der Spek, 2000; Hijma et al., 2009). Back-barrier intertidal flat deposits, dissected by tidal channels, form the upper part of the fluvio-deltaic succession of the distal delta. The sandy infill of these channels may be over 25 m thick.

## 2.3 Alluvial architecture of the proximal and central Rhine-Meuse delta

Gouw (2008, 2007) studied the upstream half of Rhine-Meuse fluvio-deltaic wedge (fig. 2) and highlighted two aspects of

alluvial architecture: (1) geometry of channel-belt sand bodies; and (2) spatial and temporal trends in alluvial architecture.

*Geometry of channel-belt sand bodies*

The geometry of a channel-belt sand body is usually characterised by the ratio between its width (SBW) and thickness (SBT): the sand body width/thickness ratio (SBW/SBT). In this paper, the term 'channel-belt sand body' refers to a sand body formed

by a single or multiple river channels. A channel-belt sand body can either be the sand body of a single channel belt (referred



to as a 'simple sand body') or be composed of multiple interconnected (amalgamated) channel belts ('complex sand body') (fig. 4).

On the Rhine-Meuse delta plain, the width of simple channel-belt sand bodies varies between 40 m and 3200 m. Their thickness
typically ranges between 5 m and 9 m, and is 6.7 m on average. It was shown that the width/thickness ratio of simple channel-belt sand bodies may decrease by a factor of 4 to 6.5 in a downstream direction, mainly due to narrowing of the sand bodies (Gouw and Berendsen, 2007). SBW and SBT of complex channel-belt sand bodies may be significantly larger than those of simple channel-belt sand bodies (fig. 4). SBW of complex channel-belt sand bodies varies between 1400 m and 4300 m; SBT ranges from 5.2 m to 10 m (Gouw, 2008). As with simple channel-belt sand bodies, SBW/SBT of complex channel-belt sand
bodies decreases downstream.

*Previously established spatial and temporal trends in alluvial architecture for the proximal and central delta*
Gouw (2008, 2007) found distinct spatial trends in the alluvial architecture of the proximal and central Rhine-Meuse delta: (1) the proportion of channel deposits (CDP) decreases in a downstream direction; (2) concurrent to CDP, the connectedness
between channel-belt sand bodies (CR) also decreases; (3) contrary to CDP, the organics proportion (OP) increases in a downstream direction; and (4) the proportion of overbank deposits (ODP) is more or less constant throughout the proximal and central delta plain. These spatial trends in alluvial architecture are attributed to variations in available accommodation space, channel-belt sand body size and flood-plain geometry (2008, 2007). For instance, CDP and CR are relatively high where floodplain width is limited.


It was found that the alluvial architecture of the central delta also varies with the age of the succession. CDP and CR for the succession formed before 3 ka BP appear to be lower than for the post-3 ka BP succession. These temporal variations in alluvial architecture are mainly related to changing sand-body geometry, because channel-belt sand bodies in the post-3 ka BP succession are significantly wider than those in the pre-3 ka BP succession. Furthermore, the interaction between aggradation
rate and avulsion frequency may have influenced alluvial architecture: CDP appears to be higher during periods of high local (i.e. natural levee) to regional (i.e. floodbasin) aggradation rates and high avulsion frequencies (Gouw, 2008; Stouthamer et al., 2011).

The architectural trends described above are valid for the proximal and central Rhine-Meuse delta. In this study, new data from
the distal delta is incorporated in order to extend our knowledge of the alluvial architecture of the preserved Rhine-Meuse fluvio-deltaic wedge, including the increasing influence of backwater morphodynamics.



## 3 Methods

Eight cross-valley geological sections (designated A-H; fig. 1) were used to determine alluvial architecture. These cross-sections were previously published by Gouw and Erkens (2007) and Hijma et al. (2009), whereby the Hijma et al. (2009)
sections were slightly extended to capture the Holocene fluvio-deltaic wedge for as much as possible. The sections were constructed with approximately 2800 borings, 724 cone penetration tests (CPTs), 278 [14]C-dates and 16 OSL-dates. The location of the sections was chosen such that they (1) capture the Holocene fluvio-deltaic succession for as much as possible; (2) are orientated perpendicular to the general flow direction which is towards the west; and (3) are distributed evenly over the study area for as much as possible. Borehole spacing along the cross-sections is ~100 m on average. For details on the applied
research methods and acquired field data, the authors refer to Gouw and Erkens (2007) and Hijma et al. (2009).

From    the    sections,    parameters    that    characterise    alluvial    architecture    were    computed (cf. Gouw, 2008): (1) channel-belt sand body geometry; (2) alluvial-architecture proportions; and (3) connectedness ratio. For cross-sections A-E, values for these alluvial-architecture parameters have already been published (Gouw, 2008, 2007) and
were slightly updated for this study. The architectural parameters for cross-sections F-H are new calculations. The original cross-sections were converted to include four basic units: (1) fluvial channel-belt deposits; (2) fluvial overbank deposits; (3) organics; and (4) intertidal (back-barrier) deposits. The units were further subdivided into sands and fines after which the alluvial-architecture parameters were calculated as described below.

### 3.1 Channel-belt sand body geometry

The method of Gouw (2008) was followed to determine the width and thickness of each channel-belt sand body in the eight cross-sections (fig. 4). Channel-belt sand body width could be readily determined within 100 m for relatively wide (≥250 m) channel-belt sand bodies and within 50 m for narrow ones (≤100 m). Data on channel-belt sand body thickness was relatively scarce, though. Exact thickness data was established for 40% of the sand bodies crossed. When channel-belt sand body thickness was unknown, averages from Gouw and Berendsen (2007) were used as a substitute. They found a thickness of
$6.7\pm1.5$ m (average$\pm1\sigma$) for individual Rhine channel-belt sand bodies in the proximal and central delta. Comparison with available field-data showed that these values are realistic for the distal delta, too (fig. 3*C*). The $1\sigma$-values were applied as margins of error in the calculations of the alluvial-architecture proportions.

### 3.2 Alluvial-architecture proportions

For all cross-sections, the proportion of fluvial channel deposits (CDP), overbank deposits (ODP) and organics (OP) were
determined. In the distal delta, fluvial channel-belt sand bodies merge into estuarine sand bodies (Hijma et al., 2009). The fluvial channel-belt sands and estuarine sands (notably bay-head delta deposits; Hijma et al., 2009) in cross-section H were therefore lumped together to calculate CDP. Additionally, the proportion of intertidal deposits (IDP) was calculated for the





sections in the distal delta. Proportion values were calculated relative to the total area of the Holocene fluvio-deltaic succession in cross-section (cf. Mackey and Bridge, 1995). For example, ODP is the cross-sectional area of all overbank deposits divided

by the total cross-sectional area. An ODP of 0.40 implies that 40% of the succession consists of overbank deposits.

The total sand proportion of each section was also calculated. Total sand proportion is defined as the sum of fluvial channel-belt sands, sands in crevasse splays and lake deposits ('coarse-grained overbank deposits' cf. Bos and Stouthamer, 2011), estuarine sands and sandy intertidal deposits ('tidal channel deposits' of Hijma et al., 2009). The proportions of these sands are calculated similar to the other alluvial-architecture parameters as described above.

### 3.3 Connectedness ratio

The connectedness ratio is the summed length of horizontal contact between channel-belt sand bodies divided by the summed width of all channel-belt sand bodies in cross-section (Gouw, 2008; cf. Mackey and Bridge, 1995) (fig. 4). The connectedness ratio is given as a fraction. For example, a CR of 0.50 implies that half the sand body width is connected to another channel-belt sand body.

### 4 Results

#### 4.1 Channel-belt sand body geometry

Geometric data for the channel-belt sand bodies in the cross-sections are summarised in table 2 and figure 5. Average channel-belt sand body width decreases in a downstream direction, from more than 1000 m upstream to *ca.* 500 m downstream (tab. 2). The exceptional high value (2823 m) for cross-section A is due to the presence of three relatively wide sand bodies in that

section (Gouw, 2008). Average channel-belt sand body thickness is practically constant throughout the study area (7-8.5 m) (tab. 2).

The downstream-decreasing trend in channel-belt sand body width/thickness ratio, as established for the proximal and central Rhine-Meuse delta (Gouw, 2008), more or less continues in the distal delta (fig. 5). The highest SBW/SBT averages (344 in

section A and 108 in section B) are found in the proximal delta, whereas values of less than 70 are found downstream. In other words, average SBW/SBT in the proximal delta is up to 5 times higher than average SBW/SBT in the distal delta. This downstream-decreasing trend is attributed to the decrease in channel-belt sand body width, because sand body thickness is near-constant over the study area (table 2).

#### 4.2 Alluvial-architecture proportions

The alluvial-architecture proportions show prominent spatial trends (fig. 6). First, the proportion of fluvial channel-belt deposits (CDP) strongly decreases in a downstream direction. It measures ~0.70 in the proximal delta (cf. Gouw, 2008) and diminishes to 0.04 downstream. This implies that the amount of fluvial channel-belt sands in the distal delta is just 5% of the





amount in the proximal delta. In the succession of the distal delta, however, fluvial channel-belt sands blend seamlessly into estuarine bay-head sands (Hijma et al., 2009). Taking this into account, CDP including estuarine sands is ~0.15 in the
downstream-most cross-section (fig. 6; table 3).

The proportion of overbank deposits (ODP) is more or less constant (0.4) in the larger part of the study area (cf. Gouw, 2008; Bos, 2010). The graph for the organics proportion (OP) reveals a distinct peak (0.30) in the transition zone from the central to the distal delta. From this point downstream, OP clearly decreases to 0.14. This trend corresponds to the appearance of intertidal
deposits in the distal delta succession. The proportion of intertidal back-barrier deposits (IDP) rapidly increases coastward to a maximum of 0.58 in the downstream-most section.

### 4.3 Total sand proportion

The total sand proportion varies between 0.18 and 0.70 (table 3) and is 0.41 for the wedge as a whole. In other words, 41% of the Rhine-Meuse fluvio-deltaic wedge is composed of sand. The total sand proportion generally decreases with downstream
distance from the delta apex, with the lowest value in cross-section G (fig. 7, table 3). This trend is reversed in the downstream-most cross-section where intertidal back-barrier and estuarine sands dominate. Here, total sand proportion amounts to 0.35.

Our data demonstrates that, except for the near-coastal area, the bulk of the sand is fluvial channel-belt sand, despite its decreasing proportion relative to the total Holocene succession (table 3). In the proximal and central delta, fluvial channel-belt
sands constitute 75-95% of the total sand proportion, whereas in the distal delta it drops to approximately 10%. Overall, fluvial channel-belt sands form 87% of all sands in the wedge. The contribution of fluvial overbank sands to the total sand proportion is relatively constant and ranges between 4.3% and 11.1% (6% overall). Estuarine and intertidal sands dominate in the downstream-most section H where they form 28.6% and 54.3%, respectively, of the total sand proportion (table 3).

### 4.4 Connectedness

The connectedness ratio (CR) of fluvial channel-belt sand bodies is roughly three times higher in the proximal delta than in the distal delta (fig. 8). CR is calculated at ~0.25 in cross-sections A-C and ~0.08 in sections D-H, with a minimum of 0.03 (cross-section F) and a maximum of 0.30 (cross-section C).

Channel-belt sand body interconnectedness increases with the proportion of channel-belt sands in the succession (fig. 8), albeit
it is not a positive (curvilinear) correlation as suggested in modelling studies (Bridge and Mackey, 1993a; Mackey and Bridge, 1995). Our data shows that CR instantly doubles as CDP rises above 0.5. Two CR populations divided by the CDP=0.5 vertical are therefore recognised: CR≥0.2 for CDP≥0.5 and CR≤0.1 for CDP<0.5. This is in accordance with previously published data (Gouw, 2008).





All Holocene channel-belt sand bodies in the proximal delta are connected to the sandy Early Holocene – Pleistocene (lowstand) substrate. In the central delta, 75-80% of the channel-belt sand bodies make contact. This figure rises in the distal delta, where approximately 90% of the Holocene channel-belt sand bodies are connected to the Early Holocene – Pleistocene substrate. This is due to the fact that most fluvial channel-belt sand bodies in the distal delta are encountered in the lower half of the Holocene succession (see figure 3C).

**5 Discussion**

Because of the importance of channel-belt sands for the alluvial architecture of fluvio-deltaic successions the discussion is focused on: (1) proposing an empirical function that describes the delta-wide trend in the proportion of channel-belt sands, based on our dataset; and (2) assessing the drivers that determine alluvial architecture in fluvial-deltaic successions.

**5.1 Delta-wide spatial trend in the proportion of channel-belt deposits**

A strong inverse relationship between CDP and downstream distance from the delta apex was found (fig. 9). It thus seems that the proportion of channel-belt sands can be estimated using the distance from the delta apex. To assess whether this could also hold for other deltas, the key factors driving the relationship should be understood. Below, the two variables of CDP – channel-belt sand body size and size of the Holocene fluvio-deltaic wedge (Bridge and Mackey, 1993a; Mackey and Bridge, 1995) – are unraveled to identify key factors for the relationship between CDP and distance from the delta apex. As a help, a causal

loop diagram is presented to visualize the interrelations between factors involved (fig. 10). Because many of these factors have been extensively elaborated in previous publications, e.g., Cohen et al. (2002), Gouw and Berendsen (2007), Gouw and Erkens (2007) and Stouthamer et al. (2011), this discussion concentrates on subjects that in our opinion have been overlooked or insufficiently highlighted before, specifically channel-belt longevity, tidal influence and the ratio between channel-belt sand body width and floodplain width.


*Variable 1: channel-belt sand body size*

Field studies (e.g., Bridge et al., 2000; Gouw, 2008) demonstrated that channel-belt sand body size strongly controls the proportion of channel-belt sands in the succession. Channel-belt sand body size is related to three variables: lateral migration rate of formative channels, channel-belt longevity, and channel size (fig. 10). Geometrically, channel-belt sand body size is

largely determined by its width because sand body width is far larger than sand body thickness (Bridge et al., 2000; Gibling, 2006; Gouw and Berendsen, 2007). Sand body width increases with lateral migration rate and longevity of the formative channel. Initially, Gouw and Berendsen (2007) attributed the high width/thickness ratios in the upper Rhine-Meuse delta (see figure 5) to high lateral migration rates of channels only. They hypothesized that these high lateral migration rates were caused by high subsoil erodibility (sandy subsoil) and high stream power (high channel gradient). They discussed that both bank

erodibility and stream power decrease in a downstream direction – the former because of an increase of erosion-resistant bank





material in the subsoil (thick layers of massive clay and peat; see figure 4*B*), the latter mainly because of decreasing river gradients in the backwater length (e.g. Blum et al., 2013) – which reduces channel lateral migrations rates and thereby sand body width, width/thickness ratios and CDP (see figures 5 and 6).

Although lateral migration rates do play a major role, Gouw and Berendsen (2007) overlooked channel-belt longevity, the second factor influencing channel-belt sand body size (fig. 10). There is general consensus (as illustrated in Karssenberg and Bridge, 2008, for example) that high channel-belt longevity favours wide sand bodies because the formative channel has had more time to widen its associated sand body. Although not yet satisfactorily evidenced by quantitative data, it seems that this theory is indeed applicable to the Rhine-Meuse delta. This is probably best exemplified in the central delta, where channel

belts with the longest period of activity are generally the widest, despite the fact that many are encased in cohesive deposits (see figure 3B). The lifespan of the formative channels was apparently of sufficient length to enable widening of their associated sand bodies, although channel migration would have been hampered by cohesive banks. This suggests that the factor 'channel-belt longevity' overpowers the factor 'subsoil erodibility' for channel-belt sand body width, at least in the central delta. This effect is probably also in place in the proximal delta, although it is less obvious due to easily erodible sediments

(sand) in the shallow subsurface favouring high channel migration rates. Most channel belts in the proximal delta have been active for several thousands of years (Berendsen and Stouthamer, 2000; Berendsen et al., 2007; Cohen et al., 2012), which is relatively long as compared to the delta-average (ca. 1000 years; Berendsen and Stouthamer, 2002). The channels thus had abundant time to form wide sand bodies, amplified by the easily erodible subsoil (see fig. 3*A*), which explains why the channel-belt sand bodies in the proximal fluvial delta are significantly wider (~900 m on average) than the delta-average (~550 m). In

short, it seems that there is a positive correlation between channel-belt longevity and channel-belt sand body width – hence width/thickness ratios and CDP – in our study area.

Gouw and Berendsen (2007) argued that in the proximal and central delta subsoil erodibility is probably dominant over stream power in explaining the downstream decrease in lateral migration, and thereby sand body width. It is suspected that the situation

in the distal delta is more complicated because of the interplay between fluvial and tidal processes (cf. Dalrymple and Choi, 2007) and in the impact of backwater effects (Blum et al., 2013). In the fluvial-dominated (upstream) part of the distal delta, fluvial-channel gradients as well as discharge per river channel, the two components of stream power, are both low. Channel gradients approach zero whereas discharge is divided over multiple river courses causing discharge per river course to be minimal. The resultant stream power per river channel is therefore extremely low leading to little lateral migration and,

consequently, narrow sand bodies. Also, as the energy to transport sediment decreases, the average grain size of the fluvial sand bodies decreases in concordance and suspended sediment concentrations rise. The latter reach a maximum somewhere in the fluvial-marine transition zone (in our study area, this would be around the x=100 coordinate). Further downstream in the distal delta, tidal currents become increasingly important. The tidal currents start to take over the role of transporting sediment which leads to a drop in suspended sediment rates and increase in average grain size (e.g., Dalrymple and Choi, 2007).





Moreover, tidal water fluxes through the channels increase seaward, i.e. discharge per channel increases due the influx of tidal waters. The increased discharge causes channel enlargement and rising stream power which likely leads to *increased* sand body width. Additionally, bank erodibility probably increases in the distal delta due to a decrease of erosion-resistant peat layers in the subsoil (see figure 6, OP-graph) in favour of immature subaqueous silt-laminated tidal clays and silty-sandy bay-head delta sediments that are easily erodible (Hijma and Cohen, 2011; Hijma et al., 2009). The presence of these soft clays

and silts in the subsoil in combination with a decrease of resistant peat expectedly caused a decrease in bank resistance or, in other words, an increase in bank erodibility. This favours lateral migration and thereby sand body width. In short, based on the above, it can be expected that the proportion of channel-belt sands decreases downstream in the fluvial-dominated part of deltas, reaching a minimum in the fluvial-marine transition zone, and then stabilises or even increases again in a seaward direction. This is exactly what is found in the data of this study (see figure 6A).


Channel size is the third factor influencing channel-belt sand body dimensions (fig. 10). Channel size certainly influences sand body size because channel depth determines minimum channel-belt sand body thickness and channel width is a minimum for channel-belt sand body width (see Gouw and Berendsen, 2007) provided that the channel is filled in with sandy bar deposits. Whereas channel-belt sand body thickness roughly reflects channel depth, channel-belt sand bodies are mostly far wider than

the width of its associated channel (Allen, 1965; Fisk, 1944; Bridge, 2003). Gouw and Berendsen (2007), for example, reported sand body width to channel width ratios of 6 to 10 for the modern Rhine and Meuse distributaries. In addition, they found significant variation in the ratio between sand body width and channel width. In other words, although channel width can be considered as a minimum for sand body width, it has a weak relation with the resultant sand body size. Therefore, channel size is not regarded by us as a key factor for channel-belt sand body size, thus CDP.


*Variable 2: dimensions of the Holocene fluvio-deltaic wedge*

The dimensions of the fluvio-deltaic wedge influences CDP because its cross-sectional area is the denominator in the CDP-calculations (cf. Mackey and Bridge, 1995). The Rhine-Meuse delta is a typical example of a fluvial system experiencing relative base-level rise and increasing aggradation rates in a downstream direction (Cohen, 2005; Van Dijk et al., 1991; Cohen

et al., 2005). As a result, the preserved Holocene fluvio-deltaic wedge thickens seaward (see figure 2). Because the wedge also widens, its volume increases strongly in a downstream direction. This geometry is typical for the Rhine-Meuse delta and it should be noted that the geometry – and architectural patterns – of fluvio-deltaic wedges varies, for example, with the direction of basin subsidence (fore-tilted versus back-tilted basins) (e.g., Heller and Paola, 1996). Two factors with regard to the geometry of the Rhine-Meuse fluvio-deltaic wedge are briefly discussed: (1) creation of accommodation space which provides

room, mainly for vertical expansion (thickening); and (2) the inherited floodplain topography influencing horizontal expansion (widening), mainly (fig. 10).





A principal prerequisite for the preservation of any fluvial succession is the availability of accommodation space, which is defined as the available space to store sediments (Blum and Törnqvist, 2000). Accommodation space in the Rhine-Meuse delta

has been created by relative base-level rise which in turn is driven by two key processes: true sea-level rise and land subsidence (Cohen, 2005; Gouw and Erkens, 2007; Hijma et al., 2009; for elaborate discussions, see Van Dijk et al., 1991) (fig. 10). The rate at which accommodation space was created in the Rhine-Meuse delta increased in a downstream direction (Cohen, 2005), as it does in most deltas with an apex in the basin hinge zone and the depocenter basin-ward. This enabled vertical expansion of the wedge and downstream thickening of the preserved fluvio-deltaic succession. In the central delta, creation of

accommodation space outpaced sediment delivery causing an increase in organics within the succession (see figure 6C). In the distal delta, the rate of accommodation space creation was even larger, but the space was mainly filled in with intertidal and estuarine muds behind a coastal barrier which formed from ~7.5 ka BP onwards. Actually, the end position of this coastal barrier also determined the dimensions of the preserved fluvio-deltaic wedge as it forms its downstream end (Hijma et al., 2009; Hijma et al., 2010; Hijma and Cohen, 2011).


The inherited floodplain topography affected the dimensions of the Rhine-Meuse fluvio-deltaic wedge because bordering Pleistocene uplands in the proximal delta (fig. 1) limit floodplain width. These uplands are absent further downstream which made horizontal expansion of the wedge possible. Because the amount of created accommodation space in the central and distal delta was indeed sufficient to enable enlargement of the wedge, the cross-sectional area of the fluvio-deltaic wedge

increases downstream. The downstream enlargement of the wedge yields relatively low CDP values, as observed in the field data (fig. 6A).

## 5.2 Relationship between CDP and normalised channel-belt sand body width

Several authors have postulated that variations in CDP should not be explained by changes in channel-belt sand body size or

floodplain size alone, but rather by variations in the ratio between these variables. The rationale behind this hypothesis is that if channel-belt sand bodies are large relative to floodplain size, they occupy a relatively large part of the available room on the floodplain – and its subsurface – which naturally leads to a high CDP of the resultant succession. Early modelling studies revealed the importance of the ratio between channel-belt sand body width (w) and floodplain width (W), w/W or 'normalised channel-belt sand body width' cf. Bridge and Mackey (1993a), for alluvial architecture (Bridge and Mackey, 1993a, b; Leeder,

1978; Bridge and Leeder, 1979; Bridge, 1999). Available field data seem to corroborate the model output despite the simplicity of these models. For example, Gouw and Autin (2008), in their field study on the Holocene alluvial architecture of the Lower Mississippi Valley (USA), indeed found that CDP increases with w/W (fig. 11). Their dataset is of limited size, though, which causes significant uncertainty in the outcome. However, Gouw (2008) also recognized a positive correlation between w/W and CDP for the fluvial-dominated part of the Rhine-Meuse delta, but the exact nature of the correlation remained obscure. To

resolve the suspected relationship between the proportion of channel-belt sands and normalised channel-belt sand body width,





new data from this study was incorporated and plotted against w/W (fig. 11). The graphic shows that CDP is extremely sensitive for changes in w/W when values of w/W are low (for our dataset, w/W<0.02). In the Rhine-Meuse delta, low w/W values are applicable to the central and distal delta due to the presence of relatively narrow channel-belt sand bodies (low w) on a wide floodplain (high W). The sensitivity of CDP for changes in w/W weakens as w/W rises (fig. 11). Relatively high

w/W values occur in the proximal Rhine-Meuse delta, caused by wide channel-belt sand bodies (high w) and limited floodplain width (low W). These results imply that the sensitivity of CDP for changes in normalised channel-belt sand body width varies spatially, and is especially strong in the central and distal delta. Because a similar trend in the relationship between CDP and w/W has been observed for the Lower Mississippi Valley (fig. 11), it is likely that spatial variations in the sensitivity of CDP on normalised channel-belt sand body width are applicable to other deltas as well. It is therefore suggested to account for these

spatial variations when explaining the alluvial architecture of fluvio-deltaic successions.

**5.3 Key factors for the CDP-trend and empirical formula**

Four factors were identified that are probably of key importance for the inverse relationship between CDP and downstream distance from the delta apex (fig. 10): channel lateral migration rate, channel-belt longevity, creation of accommodation and inherited flood-plain width. Because these factors are rather generic to fluvial systems at continental margins (see, e.g., Blum

and Törnqvist, 2000; Saucier, 1994; Bridge, 2003; Gouw, 2007; Blum et al., 2013), it is likely that the inferred relationship is applicable to other deltas as well. Indeed, available field data from the modern lower Mississippi River point to an inverse relationship of CDP with downstream distance (Gouw and Autin, 2008). Fragmentary data from ancient fluvio-deltaic deposits also suggest downstream-decreasing CDP-values (Foix et al., 2013; Klausen et al., 2014). Besides, an inverse relationship makes sense because fluvial and estuarine channel-belt sands should eventually dissipate in the marine realm. For these

reasons, it is hypothesized that downstream-decreasing CDP is probably a common characteristic of fluvio-deltaic successions. To model the spatial variability of CDP, an inverse linear function is proposed, with CDP~0.9 in the delta apex region (fig. 9). In formula:

$$CDP = 0.865 - 0.875(d/D)$$


where:

CDP = channel deposit proportion (-)

d = downstream distance from delta apex (km)

D = total distance between delta apex and shoreline (km)

d/D = relative distance downstream from delta apex (-)

Because CDP cannot be negative, minimum CDP (zero) is reached at d/D=0.99. In other words, the formula predicts fluvial and estuarine sands to be dissipated in the resultant succession at the (highstand) coastline. However, it is well possible that





the downstream-most data points in figure 9 represent minimum CDP, suggesting that CDP stabilises at ~0.1. This situation
could occur when the oldest fluvial sediments of the wedge are preserved. This is often the case in the distal part of deltas
where preservation potential of older sediments is relatively high as these are below the scouring depth of younger channels
(see, for example, figure 2).

### 5.4 Possible applications for alluvial-architecture research

In theory, the derived relationship between CDP and distance from the delta apex (or shoreline) offers opportunities for
application in alluvial-architecture research in two directions. First, provided that the palaeogeographic location within the
former delta is known, the CDP of a succession can be estimated even if the deposits are only partly exposed as is often the
case for ancient formations. For example, suppose it is assessed that a given fluvio-deltaic succession in outcrop is formed
approximately halfway between the apex of the palaeo-delta and its associated shoreline (d/D=0.5). One can then argue that
the CDP of the surveyed succession in vicinity of the outcrop location would be in the order of 0.4. The other way around, the
formula can be used to predict the palaeogeographic location based on a known value of CDP. This application is somewhat
tricky because CDP can vary significantly, even at short distances. However, the CDP of extensive outcrops or cross-sections
could be used to get a rough estimate, at least, of where the succession was formed in the palaeo-delta. A succession with a
CDP of 0.2, for instance, would indicate that the succession was formed at a relative distance of ~0.75, which is halfway
between the central delta and the shoreline, well in the distal delta.

The above-stated formula can thus be applied in two ways: to estimate sand proportions with fragmentary geologic information
and/or to understand deltaic palaeogeography. Both applications, if necessary combined with other techniques such as
shoreline trajectory analysis within sequence stratigraphic research (Bullimore and Helland-Hansen, 2009), are valuable
because they can contribute to a better interpretation of (partly exposed) fluvio-deltaic successions and optimization of research
strategies. However, the formula is derived solely from the Rhine-Meuse data set. To test whether the formula holds for other
deltas as well, field data (notably spatial trends in sand proportion) from other fluvio-deltaic settings – both modern and ancient
– should be gathered. Relatively well-studied modern and/or ancient fluvio-deltaic successions could be used as a starting
point. Possible examples are the Holocene Lower Mississippi Valley, USA (see, amongst others, Autin et al., 1991; Saucier,
1994) and extensively studied hydrocarbon reservoirs such as the Middle Jurassic Oseberg Field in the Norwegian North Sea
(Ryseth, 2000; Ryseth et al., 1998) and the Upper Carboniferous Coevorden Field in The Netherlands (see Kombrink et al.,
2007). Data from these settings would be greatly beneficial to better understand the spatial variability of channel-belt sands in
fluvio-deltaic successions.



## 6. CONCLUSIONS

1.   Fluvial sand body size in the Holocene Rhine-Meuse delta strongly decreases in a downstream direction. Average sand body width/thickness ratio (SBW/SBT) in the upper fluvio-deltaic plain is up to 5 times higher than average SBW/SBT in the lower fluvio-deltaic plain. SBW/SBT values of up to 300 are found in the upper delta, whereas they are less than 70 are in the lower delta. This trend is fully associated to a downstream-decrease in sand body width, because sand body thickness is near-constant in the Rhine-Meuse delta.

2.   A significant downstream-decrease in the proportion of fluvial channel-belt sands (CDP) in the Rhine-Meuse succession was found. CDP, including estuarine channel-belt sands, diminishes from 0.7 to 0.1. The proportion of overbank fines is near-constant (~0.4) throughout the delta. Organic matter proportion peaks (0.3) at the transition from the central to the distal delta. The connectedness ratio (CR) is roughly three times higher in the proximal delta (~0.25) than in the distal delta (~0.08).

3.   A linear inverse function is proposed to model the spatial variability of CDP in fluvio-deltaic settings. Our data demonstrates that CDP decreases linearly with downstream distance from the delta apex, with CDP~0.9 near the delta apex and approaching zero near the shoreline.

4.   Four key factors were identified that most likely explain the relationship between CDP and distance from the delta apex: channel lateral migration rate, channel-belt longevity, creation of accommodation and inherited flood-plain width. The

observed decrease in CDP is explained by downstream narrowing of fluvial sand bodies which in turn is partly the result of decreasing lateral migration rates of formative channels. Also, channel-belt sand body width appears to increase with channel-belt longevity. This effect is best noticeable in the proximal and central Rhine-Meuse delta, where channel belts with the longest lifespan are generally the widest. Accommodation space creation and floodplain geometry determine the size of the fluvio-deltaic wedge and thereby influences CDP, because the size of the Holocene wedge is the denominator

in CDP-calculations. Furthermore, the sensitivity of CDP for changes in the ratio between channel-belt sand body width and flood-plain width, referred to as normalised channel-belt sand body width, varies spatially which should be accounted for when explaining alluvial architecture.

5.   Based on our dataset, it is proposed that the proportion of fluvial channel-belt sands is generally a strong indicator for the total sand content of fluvio-deltaic successions. It was found that more than 90% of the sands in the proximal and central

delta are of fluvial-channel origin. Therefore, total sand content can be satisfactorily approximated by calculating the proportion of fluvial channel-belt sands alone; the contribution of overbank sands (e.g., crevasse sands) to the total sand content is limited. In the distal delta, however, the proportion of fluvial channel-belt sands alone does not reflect total sand content because a mix of fluvial channel-belt sands, estuarine sands and intertidal sands largely form the total sand proportion, with the importance of fluvial channel-belt sands rapidly diminishing coastward.




With this paper, high-resolution quantitative data and spatial trends on the alluvial architecture are available for an entire delta for the very first time. The unique parameterisations based on real-world data hopefully ignites further research on alluvial architecture in order to enhance our understanding of delta development and sediment preservation, and to improve existing fluvial stratigraphy models.

**Author contribution**

MJPG used the cross-sections to calculate alluvial-architectural parameters. The authors had equal share in the analysis and writing part of the paper.

**Competing interests**

The authors declare that they have no conflict of interest.

**Acknowledgements**

[To follow]

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



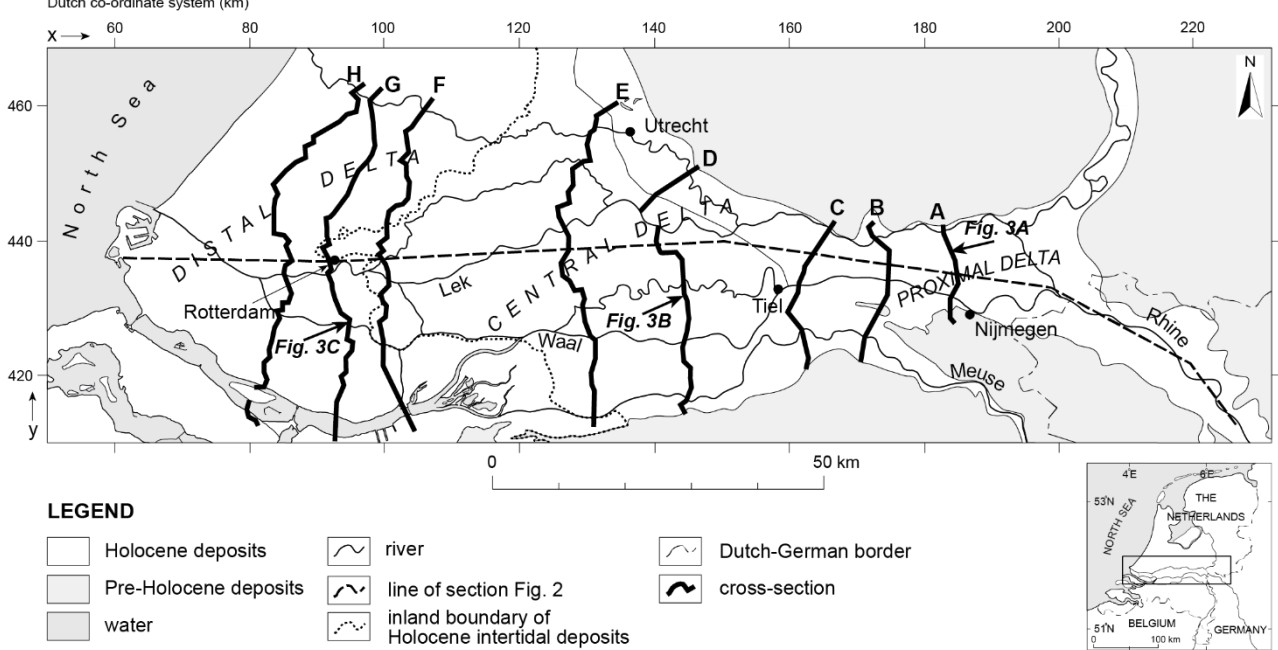

Figure 1 Gouw and Hijma

**Figure 1: Location of the Rhine-Meuse delta, The Netherlands. The cross-sections used in this study are designated A-H: cross-sections A-E are from Gouw and Erkens (2007), sections F-H are from Hijma et al. (2009, modified). Definition of proximal (x-coordinates 160-220 km), central (x=110-160 km) and distal delta (x=70-110 km) is based on Stouthamer et al. (2011). The current delta apex is located ~20 km upstream of the Dutch-German border.**

Earth **Surface**
**Dynamics**
Discussions



*Figure 2 Gouw and Hijma*

**Figure 2: Schematic longitudinal section and sequence stratigraphy of the Holocene fluvio-deltaic wedge of the Rhine-Meuse delta (Hijma and Cohen, 2011 modified). Sequence-stratigraphic classification is from Hijma and Cohen (2011) and Cohen and Hijma**
**(2014). The wedge largely consists of a stacked succession of clastic fluvial deposits and organics (peat). Towards the coast, estuarine and intertidal sedimentary lobes form the larger part of the wedge.**



*Figure 3 Gouw and Hijma*

**Figure 3: Transects showing typical fluvio-deltaic stratigraphy of the proximal (*A*), central (*B*) and distal (*C*) Holocene Rhine-Meuse delta (after Gouw and Erkens (2007) and Hijma et al. (2009 modified). See figure 1 for locations. Kilometre-scale above each transect refers to the original cross-sections. Note: [14]C-dates are given in [14]C-yr BP, OSL-dates in kyr BP. See text for a general description of the transects.**




**(A) Unconnected: simple sandbodies**

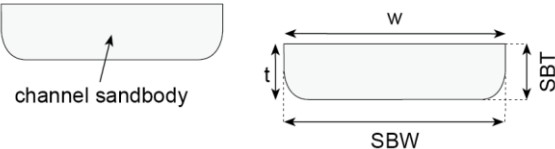

**(B) Interconnected: complex sandbodies**

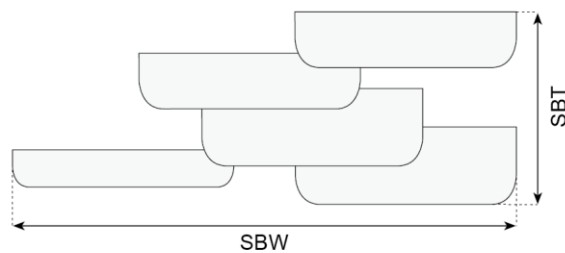

**(C) Connectedness ratio (CR)**

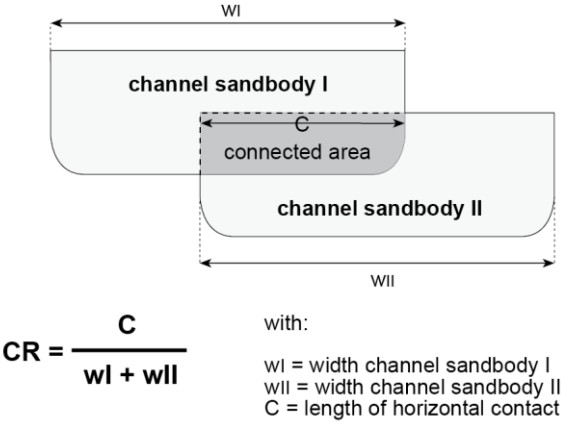

$$CR = \frac{C}{wI + wII}$$

with:

wI = width channel sandbody I
wII = width channel sandbody II
C = length of horizontal contact

*Figure 4 Gouw and Hijma*

**Figure 4: Definition diagram showing geometric properties of fluvial channel-belt sand bodies (after Gouw (2008), based on Mackey and Bridge (1995)). A channel-belt sand body may be composed of either a single channel-belt sand body (simple channel-belt sand body) or multiple interconnected channel-belt sand bodies (complex channel-belt sand body). The dimensions (SBW, SBT) of a simple channel-belt sand body are equal to the dimensions of the channel-belt sand body (w, t) that constitutes the simple sand body (*A*). In case of complex channel-belt sand bodies, SBW and SBT may be significantly larger than the dimensions of the individual channel-belt sand bodies that are part of the complex sand body (*B*). Calculation of the connectedness ratio (CR) is schematically shown in (*C*).**





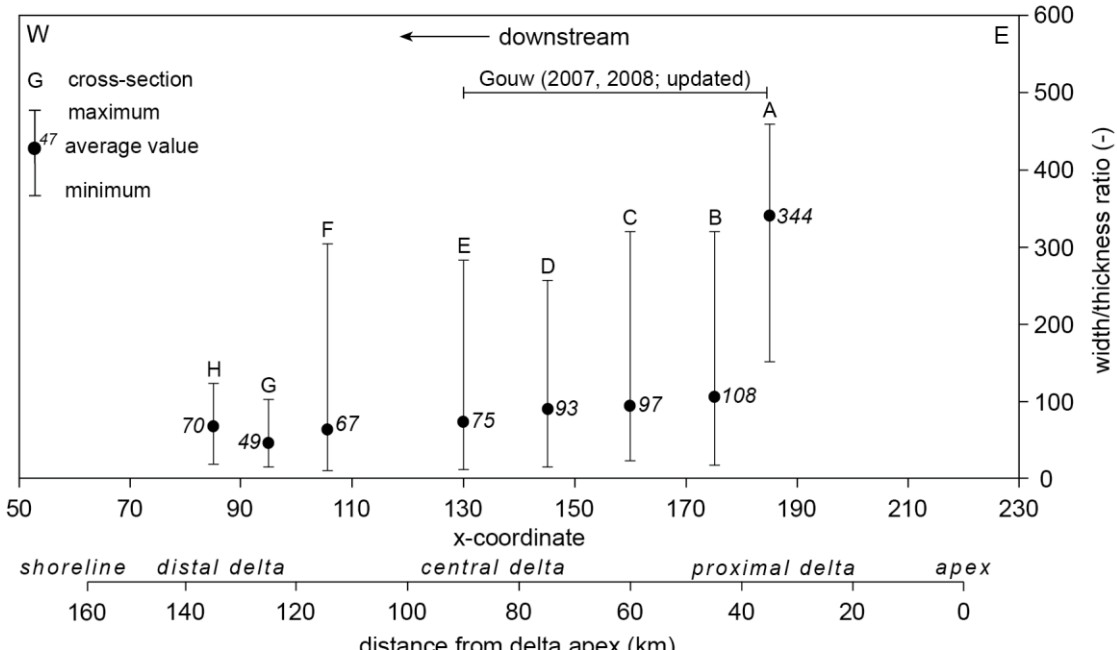

Figure 5 Gouw and Hijma

**Figure 5: Average width/thickness ratios (SBW/SBT) as calculated for each cross-section. The ranges result from applying a thickness of 5.2 m, 6.7 m, and 8.2 m (average±1σ) for channel-belt sand bodies with an unknown thickness (Gouw, 2008; cf. Gouw and Berendsen, 2007). The data records in this paper are plotted against the x-coordinate because general flow direction is towards the west which makes the x-coordinate a suitable measure for downstream distance. Average SBW/SBT decreases with downstream distance from the delta apex. Data for sections A-E is updated from Gouw (2008, 2007).**





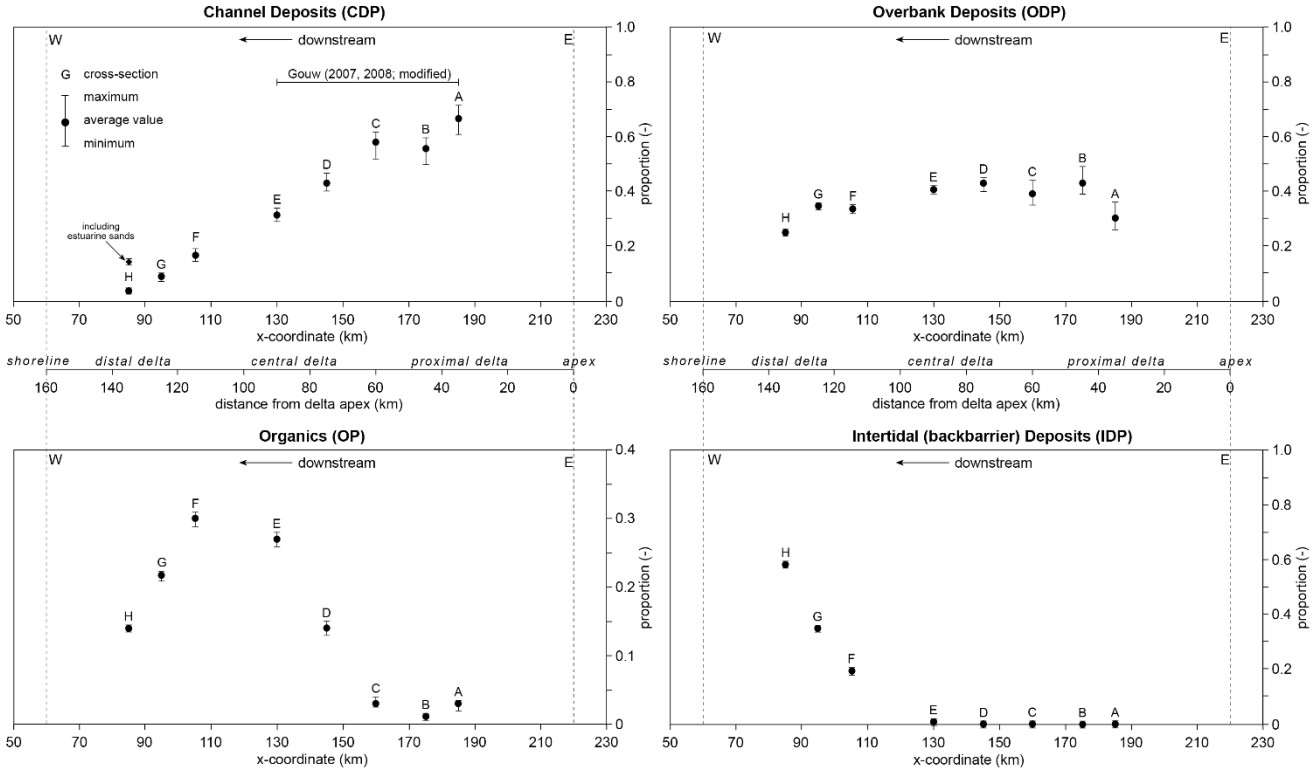

*Figure 6 Gouw and Hijma*

**Figure 6: Proportions of channel deposits (CDP), overbank deposits (ODP), organic matter (OP) and intertidal deposits (IDP) as established for the cross-sections. Ranges are as in figure 5. CDP clearly decreases in a downstream direction, whereas ODP is near-constant in the larger part of the Rhine-Meuse delta. OP peaks in the central-to-distal-delta transition zone and subsequently decreases coastward. This decrease is associated with a steep increase in the proportion of intertidal back-barrier deposits (IDP), which is 0.6 at a maximum in our study area. Data for sections A-E is slightly modified from Gouw (2008, 2007). Note different vertical scale for the OP-graph.**


Earth **Surface**
**Dynamics**
Discussions

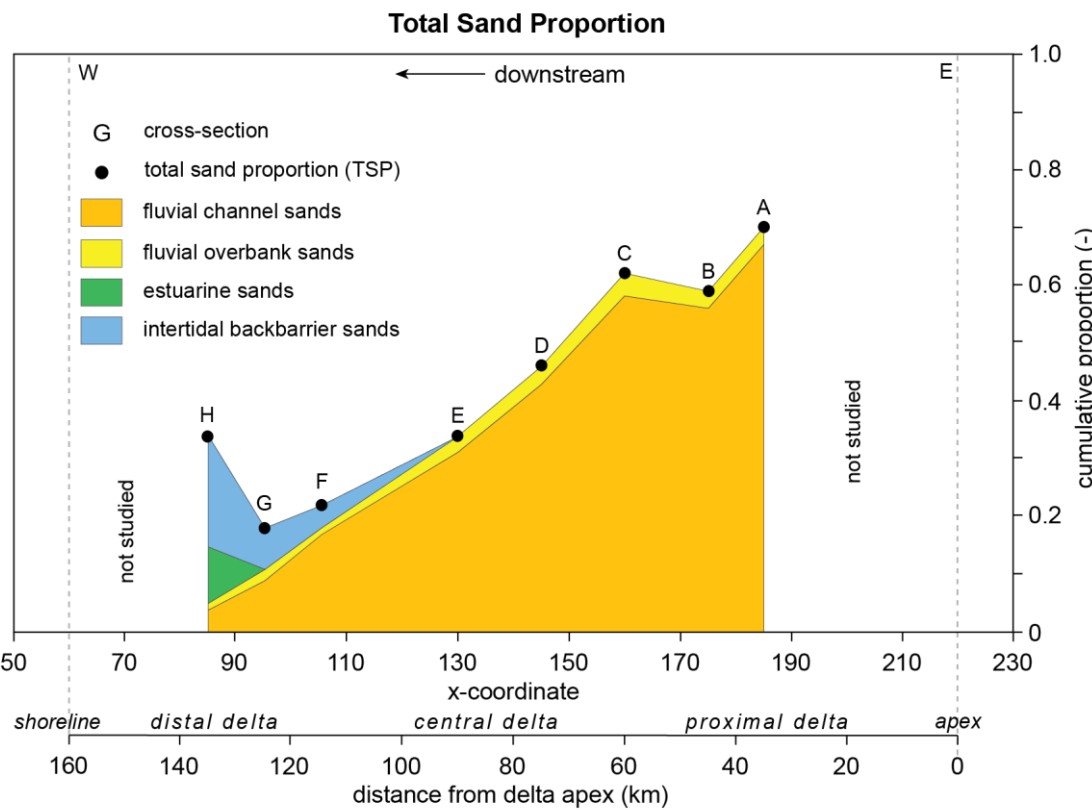


Figure 7 Gouw and Hijma

**Figure 7: Diagram showing the contributors to the total sand proportion for each cross-section. The total sand proportion is the sum of fluvial channel-belt sands, coarse-grained overbank deposits in crevasse splays and lake deposits (cf. Bos and Stouthamer, 2011), estuarine sands (notably bay-head delta deposits; Hijma et al., 2009) and sandy back-barrier intertidal deposits. In the upstream delta, most sand is stored in fluvial channel-belt sand bodies. In contrast, most sand in the downstream delta is estuarine and intertidal in nature.**






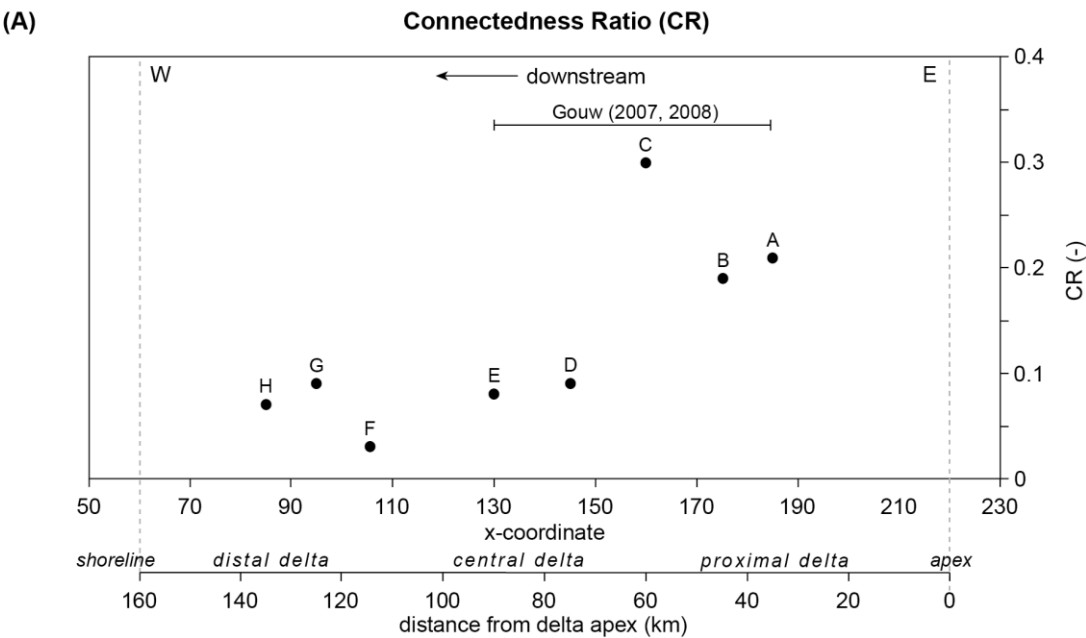

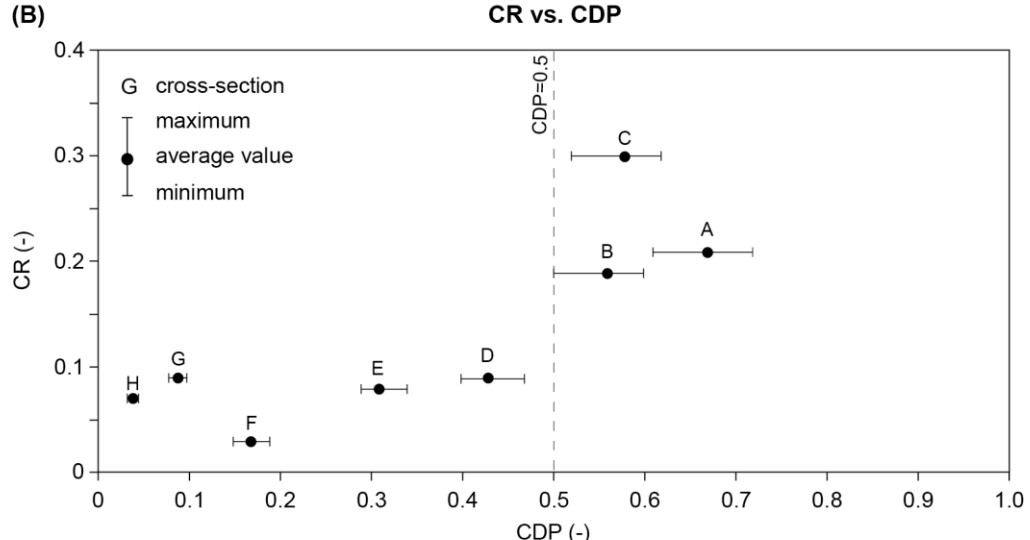

*Figure 8 Gouw and Hijma*

**Figure 8: A) CR as established for the cross-sections and (B) CR plotted against CDP. CR decreases in a downstream direction. The high value for cross-section C is due to the presence of an exceptionally large complex channel-belt sand body. CR is significantly higher when CDP exceeds 0.5 (cf. Gouw, 2008). Data for cross-sections A-E is from Gouw (2007, 2008).**






Earth **Surface**
**Dynamics**
Discussions

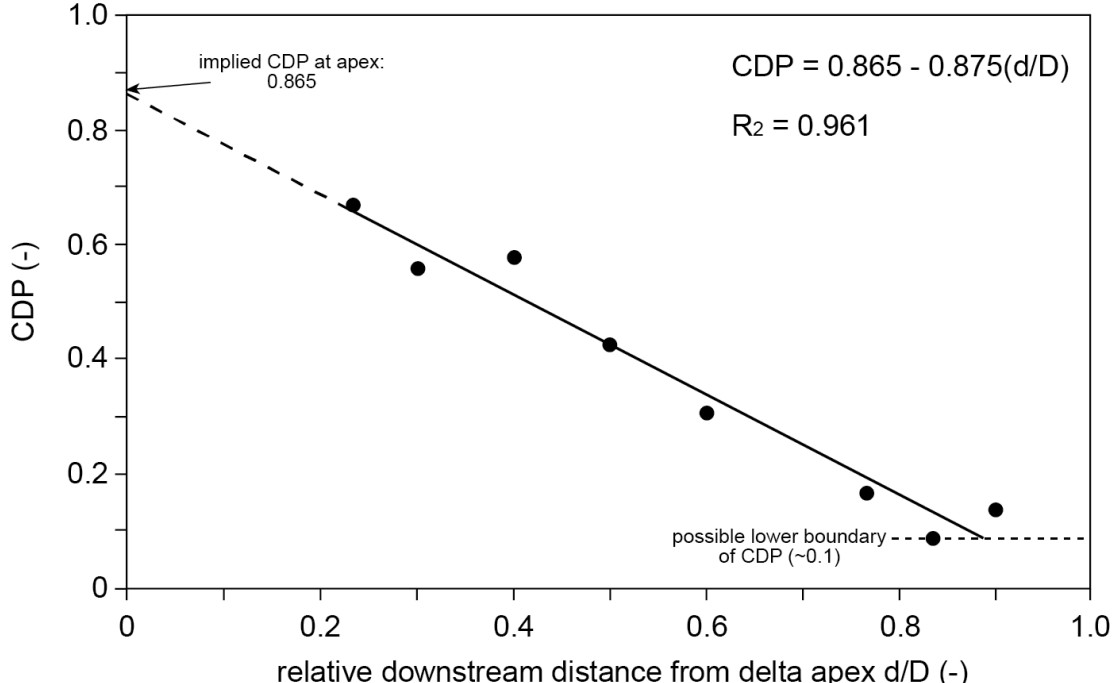

CDP = 0.865 - 0.875(d/D)

$R_2$ = 0.961

implied CDP at apex: 0.865

possible lower boundary of CDP (~0.1)

*Figure 9 Gouw and Hijma*

**Figure 9: Relationship between CDP and downstream distance from the delta apex. Distance is noted relative to the total distance between the delta apex and the shoreline (relative distance d/D, with d=downstream distance from delta apex and D=total distance between delta apex and shoreline). The derived linear function suggests a CDP of ~0.9 near the delta apex (d/D=0). CDP is at a minimum (0.10-0.15) in the distal delta (d/D>0.8).**








*Figure 10 Gouw and Hijma*

**Figure 10: Causal loop diagram displaying the interrelated variables for CDP. A plus-sign indicates a positive relationship between**
**variables (variables change in the same direction), a minus-sign indicates a negative relationship. CDP is calculated with two**
**variables (marked in green boxes): channel-belt sand body size and size of the fluvio-deltaic wedge. These in turn are largely**
**determined by channel lateral migration rate, accommodation space and inherited floodplain geometry (width and topography) for**
**which they are considered as the key driving factors (greenblue boxes) explaining the relationship between CDP and downstream**
**distance from delta apex. See text for discussion.**






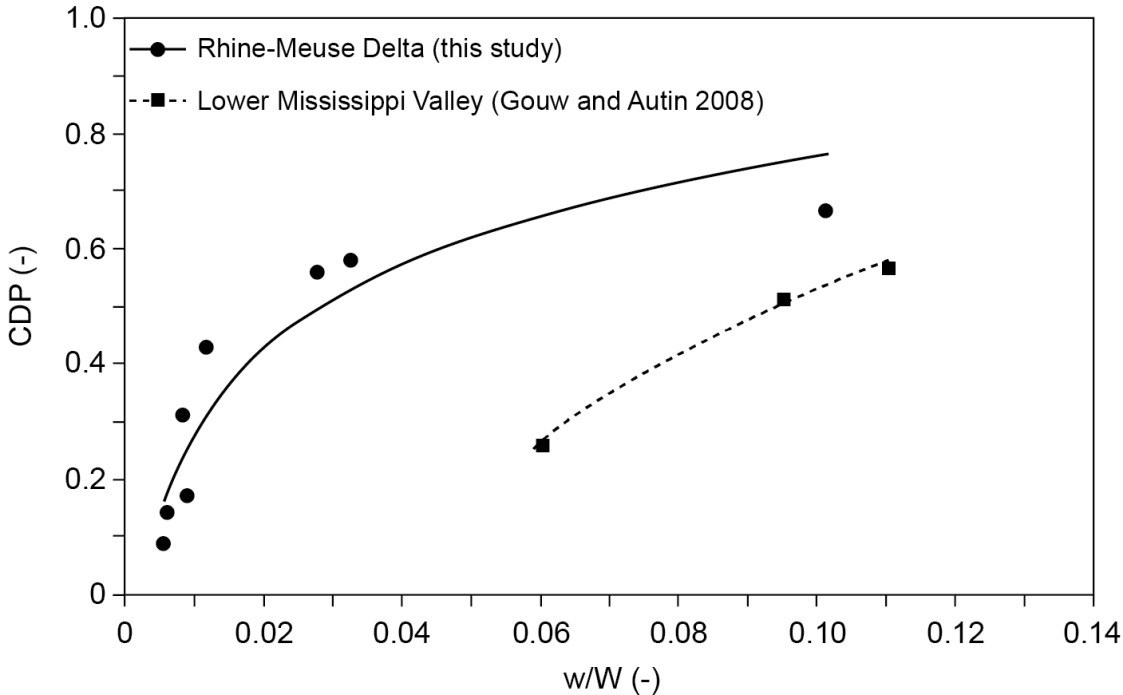

*Figure 11 Gouw and Hijma*

**Figure 11: Channel-belt deposit proportion (CDP) plotted against normalised channel-belt sand body width (w/W), Rhine-Meuse delta (points) and Lower Mississippi Valley (rectangles). The data reveals a positive relationship between the two variables. The Rhine-Meuse data clearly displays a strong increase of CDP with w/W when w/W is low (<0.02). The sensitivity of CDP for changes**

**in w/W declines with increasing w/W. Data for the Lower Mississippi Valley is from Gouw and Autin (2008).**






**Table 1: Chronostratigraphy and lithostratigraphy for the Holocene Rhine-Meuse delta. Chronostratigraphy for the Holocene following Van Geel et al. (1981), the Late Glacial following Hoek (2008) and Rasmussen et al. (2006). Lithostratigraphy nomenclature**
**is cf. Westerhoff et al. (2003).**

| CHRONOSTRATIGRAPHY | | | ¹⁴C ka BP | cal ka BP | LITHOSTRATIGRAPHY aeolian | fluvial | marine | organic |
|---|---|---|---|---|---|---|---|---|
| HOLOCENE | Late Holocene | Subatlantic | | | no deposits | ECHTELD FORMATION | NAALDWIJK FORMATION | NIEUWKOOP FORMATION |
| | | | 2.5 | 2.6 | | | | |
| | Middle Holocene | Subboreal | | | | | | |
| | | | 5.0 | 5.7 | | | | |
| | | Atlantic | | | | | | |
| | | | 7.9 | 8.7 | | | | |
| | Early Holocene | Boreal | | | | | | |
| | | | 9.15 | 10.25 | | | | |
| | | Preboreal | | | BOXTEL FORMATION | KREFTENHEYE FORMATION | no deposits | local organic beds: Wychen Member of the KREFTENHEYE FORMATION |
| | | | 10.15 | 11.65 | | | | |
| PLEISTOCENE | Weichselian | Late Weichselian (Late Glacial) | Younger Dryas | | | | | |
| | | | 10.95 | 12.85 | | | | |
| | | Allerød | | | | | | |
| | | | 11.7 | 13.95 | | | | |
| | | Older Dryas | 12.1 | 14.03 | | | | |
| | | Bølling | | | | | | |
| | | | 12.5 | 14.64 | | | | |
| | | Middle Weichselian (Pleniglacial) | | | | | | |

*Table 1 Gouw and Hijma*




**Table 2: Dimensions of the channel-belt sand bodies in the cross-sections. Average (av), minimum (min), and maximum (max) values for the channel-belt sand body dimensions are given for each section.**

| Cross-section | Number of sandbodies | Simple : complex[a] | With estimated thickness[b] | SBW (m) | | | SBT[c] (m) | | | SBW/SBT[c] (-) | | | Data source[d] |
|---|---|---|---|---|---|---|---|---|---|---|---|---|---|
| | | | | Av | Min | Max | Av | Min | Max | Av | Min | Max | |
| A | 3 | 1 : 2 | 3 | 2823 | 1385 | 4281 | 8.4 | 6.7 | 9.3 | 344 | 152 | 460 | 1, 2 |
| B | 11 | 8 : 3 | 10 | 999 | 63 | 3728 | 7.3 | 2.6 | 11.7 | 108 | 16 | 319 | 1, 2 |
| C | 13 | 9 : 4 | 13 | 923 | 163 | 4480 | 7.9 | 6.7 | 14.0 | 97 | 24 | 320 | 1, 2 |
| D | 28 | 22 : 6 | 20 | 727 | 93 | 2624 | 7.1 | 4.0 | 11.0 | 93 | 15 | 257 | 1, 2 |
| E | 31 | 25 : 6 | 13 | 582 | 70 | 2458 | 7.3 | 5.0 | 12.9 | 75 | 11 | 283 | 1, 2 |
| F | 29 | 26 : 3 | 17 | 547 | 82 | 2033 | 9.0 | 6.7 | 16.9 | 67 | 9 | 303 | 3, this paper |
| G | 27 | 24 : 3 | 9 | 366 | 98 | 1488 | 7.4 | 3.5 | 17.1 | 49 | 16 | 140 | 3, this paper |
| H | 10 | 8 : 2 | 4 | 530 | 133 | 1428 | 7.3 | 3.3 | 13.0 | 70 | 18 | 123 | 3, this paper |

[a] Number of simple sandbodies versus complex ones. For definitions, see main text.
[b] Number of sandbodies with an estimated thickness. When channel sandbody thickness was unknown, data from Gouw and Berendsen (2007) were used as an estimate. See main text for explanation.
[c] Data presented is for the case wherein a thickness of 6.7 m is taken as an estimate for channel sandbodies with an unknown base (cf. Gouw and Berendsen (2007))
[d] Data sources:
   1    Gouw and Erkens (2007)
   2    Gouw (2007, 2008) updated
   3    Hijma et al. (2009)

*Table 2 Gouw and Hijma*






**Table 3: Data on the Total Sand Proportion (TSP) and Its Contributors for the Rhine-Meuse Fluvio-deltaic Wedge. Proportions are relative to the total Holocene fluvio-deltaic succession. Percentages are relative to the total sand proportion.**

| Cross-section | Relative Distance from Delta Apex d/D (-) | Total Sand Proportion (-)[a] | Proportions Relative to Total Holocene Succession (-) | | | | Percentage of Total Sand Proportion (%) | | | |
|---|---|---|---|---|---|---|---|---|---|---|
| | | | Fluvial channel-belt sands | Fluvial overbank sands[b] | Estuarine sands[c] | Intertidal back-barrier sands[d] | Fluvial channel-belt sands | Fluvial overbank sands | Estuarine sands | Intertidal back-barrier sands |
| A | 0.23 | 0.70 | 0.67 | 0.03 | 0 | 0 | 95.7 | 4.3 | 0 | 0 |
| B | 0.30 | 0.59 | 0.56 | 0.03 | 0 | 0 | 94.9 | 5.1 | 0 | 0 |
| C | 0.40 | 0.62 | 0.58 | 0.04 | 0 | 0 | 93.5 | 6.5 | 0 | 0 |
| D | 0.50 | 0.46 | 0.43 | 0.03 | 0 | 0 | 93.1 | 6.9 | 0 | 0 |
| E | 0.60 | 0.34 | 0.31 | 0.03 | 0 | 0 | 90.4 | 9.6 | 0 | 0 |
| F | 0.77 | 0.22 | 0.17 | 0.01 | 0 | 0.04 | 76.2 | 5.8 | 0 | 17.9 |
| G | 0.83 | 0.18 | 0.09 | 0.02 | 0 | 0.07 | 50.0 | 11.1 | 0 | 38.9 |
| H | 0.90 | 0.35 | 0.04 | 0.01 | 0.10 | 0.19 | 11.4 | 5.7 | 28.6 | 54.3 |
| Total wedge | n/a | 0.41 | 0.36 | 0.03 | <0.01 | 0.02 | 86.8 | 6.2 | 1.3 | 5.7 |

[a] Total Sand Proportion is the sum of fluvial channel sands (CDP), fluvial overbank sands, estuarine sands and marine sands.
[b] Sands in crevasse splays and lake deposits ('coarse-grained overbank deposits' cf. Bos and Stouthamer 2011)
[c] Notably bay-head delta deposits (Hijma et al., 2009). Data from Bos and Stouthamer (2011)
[d] Sandy intertidal deposits, mainly tidal channel fills


*Table 3 Gouw and Hijma*