# Peer review of "From apex to shoreline: fluvio-deltaic architecture for the Holocene Rhine-Meuse delta, The Netherlands"

_Earth Surface Dynamics, 2021_

## Author Response (AR1)

Reply to comments (Gouw and Hijma, ESurf paper)

Dear editor,

We were happy with the reviews as they helped us in further improving the paper. We feel we have addressed all major remarks in an appropriate manner. In general the referees were positive about the data itself and the presentation of the data. Hardly any comments on those sections. They had significant concerns about the discussion section, so we paid extensive attention to refining our discussion section

Best,

Marc Hijma
Deltares

Reviewer words in black, reply in red

**Referee 1**

This manuscript evaluates changes in channel body architecture of the Holocene Rhine-Meuse delta. Using perhaps the highest resolution dataset of any field-scale delta stratigraphy, the authors find a remarkably linear decrease in the channel deposit proportion with distance down the delta, a threshold transition in channel connectivity ratio with distance down the delta, and a hooked trend of sand fraction with total distance down the delta that accounts for estuarine and intertidal sands. While these trends are analyzed in the context of reservoir analysis, I think they would also be interesting in a sand availability/sustainability context or the sand fraction required for deltas to aggrade with rising sea levels. These trends are clear and compelling, and I agree that the trends may be characteristic of many modern and ancient deltas.

Unfortunately, I found the context of these results to need significant improvement before it can be considered for publication. By context, I mean that the introduction makes promises that are not met, and the discussion does not yield an improved process-based understanding or challenge existing concepts. I would very much like to see these data published one day, but this context will need to be improved.

The abstract and introduction promise analysis of the Rhine-Meuse dataset in the context of backwater hydrodynamics. I was excited to read about this, because the Rhine-Meuse has rarely been studied in this context. However, this never appears, and I don't think that the backwater length of the system is ever even given. There are many recent studies that give generic predictions about channel dynamics in the backwater zone (Chadwick et al., 2019, 2020; Fernandes et al., 2016; Martin et al., 2018). The narrowing of channel bodies, in particular, appears to be characteristic of backwater zones, including the Rhine-Meuse (Fernandes et al., 2016). As stated above, the trends you show are clear, so it would be valuable to compare them to previous predictions.

We now have significantly more focus on the backwater zone in the discussion section

In the discussion, the authors attempt to explain the trends primarily as a function of channel belt sand body size, longevity, creation of accommodation, and inhereited floodplain width, and a causal loop diagram. I think this section could also be improved. For example, it is my understanding that the longevity of most channels in the Rhine-Meuse are quantified (Stouthamer and Berendsen, 2001). This suggests that a channel

belt width, channel longevity graph could be constructed and empirically analyzed. The paragraph beginning L368 argues that the loss of stream power explains the reduction in mobility. However, backwater studies have shown that shear stress may increase with proximity to the coastline (Lamb et al., 2012; Nittrouer et al., 2012; Smith et al., 2020). The claim that clays and silts are more erodible than peats is an interesting one (L384), but seems contrary to prevailing thoughts that cohesive fine sedimentation is the most resistant material in rivers and deltas (Dunne and Jerolmack, 2018; Edmonds and Slingerland, 2009). In summary, there are concepts out there to flesh out the understanding of controls, but they are not used.

In improving our discussion section we address all points above, and more, to deepen the discussion about the controls on alluvial architecture. We have added a figure showing the relation between longevity and channel-belth width

This laundry list of concerns is meant to bring the manuscript in line with the current state of understanding of fluvial deltaic channel dynamics. Some or all of these concepts from the literature may be wrong, or may not apply to the Rhine-Meuse. I would welcome results and a paper that contradict our current understanding, as the field data available to the authors should trump models and push the field forward. However, the qualitative claims made by the authors are too vague to prove the cause of the stratigraphic changes, and are frequently in contradiction with alternate explanations. Another avenue I would recommend is to focus less on a morphodynamic explanation of the trends (as the discussion does) and more on implications. It is of course up the authors, but I think the same data could be used to really advance the growing field of sand resources, with only a brief description of similarities and differences to backwater controls on channel architecture, and I would still consider it a significant advance. I hate it when reviewers talk about the paper they would have written! I just bring it up as an option because the data is fascinating, and I want it out in the world general models of the deltaic clustering of sand bodies and sands or not!

We agree that sand aggregate mining benefit from alluvial architecture studies like ours and we now make a direct link to it.

Chadwick, A. J., Lamb, M. P., Moodie, A. J., Parker, G., and Nittrouer, J. A.: Origin of a Preferential Avulsion Node on Lowland River Deltas, Geophys. Res. Lett., 46, 4267–4277, https://doi.org/10.1029/2019GL082491, 2019.

Chadwick, A. J., Lamb, M. P., and Ganti, V.: Accelerated river avulsion frequency on lowland deltas due to sea-level rise, Proc. Natl. Acad. Sci., 117, 17584–17590, https://doi.org/10.1073/pnas.1912351117, 2020.

Dunne, K. B. J. and Jerolmack, D. J.: Evidence of, and a proposed explanation for, bimodal transport states in alluvial rivers, Earth Surf. Dyn., 6, 583–594, https://doi.org/10.5194/esurf-6-583-2018, 2018.

Edmonds, D. A. and Slingerland, R. L.: Significant effect of sediment cohesion on delta morphology, Nat. Geosci., 3, 105–109, 2009.

Fernandes, A. M., Törnqvist, T. E., Straub, K. M., and Mohrig, D.: Connecting the backwater hydraulics of coastal rivers to fluvio-deltaic sedimentology and stratigraphy, Geology, 44, 979–982, https://doi.org/10.1130/G37965.1, 2016.

Lamb, M. P., Nittrouer, J. A., Mohrig, D., and Shaw, J. B.: Backwater and river plume controls on scour upstream of river mouths: Implications for fluvio-deltaic morphodynamics, J. Geophys. Res., 117, F01002, https://doi.org/10.1029/2011JF002079, 2012.

Martin, J., Fernandes, A. M., Pickering, J., Howes, N., Mann, S., and McNeil, K.: The Stratigraphically Preserved Signature of Persistent Backwater Dynamics in a Large Paleodelta System: The Mungaroo Formation, North West Shelf, Australia, J. Sediment. Res., 88, 850–872, https://doi.org/10.2110/jsr.2018.38, 2018.

Nittrouer, J. A., Shaw, J. B., Lamb, M. P., and Mohrig, D.: Spatial and temporal trends for water-flow velocity and bed-material transport in the lower Mississippi River, Geol Soc Am Bull, 124, 400–414, https://doi.org/10.1130/B30497.1, 2012.

Smith, V., Mason, J., and Mohrig, D.: Reach-scale changes in channel geometry and dynamics due to the coastal backwater effect: the lower Trinity River, Texas, Earth Surf. Process. Landf., 45, 565–573, https://doi.org/10.1002/esp.4754, 2020.

Stouthamer, E. and Berendsen, H. J. A.: Avulsion Frequency, Avulsion Duration, and Interavulsion Period of Holocene Channel Belts in the Rhine-Meuse Delta, The Netherlands, J. Sediment. Res., 71, 589–598, https://doi.org/10.1306/112100710589, 2001.

**Referee 2**

This paper describes the fluvio-deltaic architecture of the Holocene Rhine-Meuse delta including marine and estuarine deposits which were not described in literature before. In the introduction, the authors describe the need for more data for geological modelling and formulate 3 aims: describe the new architectural data including the marine & estuarine deposits, develop a new function that describes delta wide spatial trends in channel deposit proportion, and present new insides for other fluvio-deltaic settings.

My main issue with this paper is the discussion section, and how it connects to the rest of the paper. Firstly, this section starts with new objectives and therefore is not connected to the aims in the introduction. It would be much clearer for the reader if this section starts with a reminder of the aims stated in the introduction, how the structure of the discussion relates to these aims, and why subsequently the authors choose to focus on the importance of channel belt sands.

Good suggestion that we implemented directly.

Secondly, the discussion section speculates on the formative processes and thereby the key factors for the trend in CDP ("Four factors were identified that are probably of key importance for the inverse relationship between CDP and downstream distance from the delta apex (fig. 10): channel lateral migration rate, channel-belt longevity, creation of accommodation and inherited flood-plain width"), which are only introduced within the discussion section. Figure 9 and 10 show new data which are not in the results section, and most of the relations mentioned in the results section are not directly discussed here. Therefore, the whole discussion reads disconnected from the previous sections.

We have improved the discussion section significantly, addressing above points. The connection between the discussion part and the rest of the paper could indeed be strengthened, so thanks for pointing that out!

Looking at the introduction, I am missing a more detailed discussion on how this datasets provides new insights related to marine and estuarine deposits and if relations found in literature still hold, and on the significance for geological modelling. Are the found relations specific enough to aid geological modelling?

While improving the discussion section, we also addressed these points.

Furthermore, one of the aims is to present new insights for other deltas, but the discussion section specifically mentions more field data should be gathered from other deltas to be able to say whether the equation holds. It would therefore be better to remove this aim and comment on it in the discussion.

Done so.

**Specific comments:**

*Introduction*

L50: " notable exceptions": what are the conclusions of these studies? Are they relevant for the current study?

To avoid making the introduction too long, we have removed these references.

Paragraph L51-59: Do these models need specific data from the system that is modelled, or will data from e.g. the Rhine-Meuse delta also help models of other deltas significantly?

We think that data from similar system can be used in other deltas, analogues, as well.

The structure of the introduction is clear, but the link between the aims of this paper and the identified research gaps could be more specific:

-    Aim 2: with the goal to use it for geological modelling? Is the aim to develop this function for models of the Rhine-Meuse delta or also for other deltas?

Other deltas as well, added it to the aim

-    Aim 3: Is the novelty related to the marine & estuarine deposits? Or will the relations in this paper also provide new insides for the fluvial part?

The novelty lies in the backwater length of the system, added it to the aim

After reading the discussion, I think formulating these aims again and more specifically is important for the structure of the remainder of the paper.

*Section 2*:

The beginning of section 2 could use an outline of the structure of the section and how

this relates to the objectives of the paper, to make this section it easier to follow. Why is there a section on the cross-sections in between the sections on the architecture? Is the architecture described in section 2.1 and 2.3 not based on these cross-sections?

The cross-sections in 2.2 are smaller parts of the cross section used for 2.1 and 2.3. They are put here to illustrate some key features of the delta. I understand that this is not clear and added an outline to section 2.

*Section 3:*

L235: why/how were they updated?

Correcting mainly some earlier typo's and one cross-section became a bit longer and hence some values changed a bit.

L246: how does this information follow from figure 3C?

From the thickness from the channel sand bodies.

*Section 4:*

Section 4.1: the uncertainty on the width/thickness ratio is really high, for most cross-sections in the order of the mean value. The downstream decreasing trend is therefore not convincing to me, only if you compare cross-section A with H and G. Please comment on the significance of these findings and how this influences the conclusions.

The shown band width around the average is not the uncertainty (in terms of a standard deviatiob), but shows the full range between the min and max value.

**Some technical corrections:**

Throughout the manuscript, relevant information is mentioned between brackets. E.g. line 100 "(note that all dates are in calendar years, unless stated otherwise)" and line 105 "(an up to 5 m thick bed of fine sand and silty and sandy clay, mainly)". Personally, I would find it clearer and easier to read if this information is included in a more active way, so the authors might consider removing some of the brackets and rephrasing some sentences.

Good point, changed the frequency of using brackets

L57: "would be helpful": consider rephrasing to e.g. "is helpful/necessary" to make it sound like it is essential to have this information, assuming this is what you mean.

Changed it accordingly

**Citation**: https://doi.org/10.5194/esurf-2021-42-RC2